# Video Analysis of Small Bowel Capsule Endoscopy Using a Transformer Network

**DOI:** 10.3390/diagnostics13193133

**Published:** 2023-10-05

**Authors:** SangYup Oh, DongJun Oh, Dongmin Kim, Woohyuk Song, Youngbae Hwang, Namik Cho, Yun Jeong Lim

**Affiliations:** 1School of Electrical and Computer Engineering, Seoul National University, 1 Gwanak-ro, Kwanak-gu, Seoul 08826, Republic of Korea; syup5@snu.ac.kr (S.O.); whsong@snu.ac.kr (W.S.); 2Department of Internal Medicine, Dongguk University Ilsan Hospital, Dongguk University College of Medicine, Goyang 10326, Republic of Korea; mileo31@naver.com; 3JLK TOWER, Gangnam-gu, Seoul 06141, Republic of Korea; dmkim@jlkgroup.com; 4Department of Electronics Engineering, Chungbuk National University, Cheongju 28644, Republic of Korea; ybhwang@chungbuk.ac.kr

**Keywords:** artificial intelligence, transformer, capsule endoscopy, video-analysis

## Abstract

Although wireless capsule endoscopy (WCE) detects small bowel diseases effectively, it has some limitations. For example, the reading process can be time consuming due to the numerous images generated per case and the lesion detection accuracy may rely on the operators’ skills and experiences. Hence, many researchers have recently developed deep-learning-based methods to address these limitations. However, they tend to select only a portion of the images from a given WCE video and analyze each image individually. In this study, we note that more information can be extracted from the unused frames and the temporal relations of sequential frames. Specifically, to increase the accuracy of lesion detection without depending on experts’ frame selection skills, we suggest using whole video frames as the input to the deep learning system. Thus, we propose a new Transformer-architecture-based neural encoder that takes the entire video as the input, exploiting the power of the Transformer architecture to extract long-term global correlation within and between the input frames. Subsequently, we can capture the temporal context of the input frames and the attentional features within a frame. Tests on benchmark datasets of four WCE videos showed 95.1% sensitivity and 83.4% specificity. These results may significantly advance automated lesion detection techniques for WCE images.

## 1. Introduction

With the introduction of wireless capsule endoscopy (WCE) for the first time in 2000, there has been a paradigm shift in small bowel examination [1,2]. After the patient swallows the capsule, it moves along the digestive tract from the mouth to the anus without user manipulation. Indeed, WCE is essential for monitoring Crohn’s disease, small bowel bleeding, small bowel tumors, and inherited polyp syndrome, and it also has the advantage of being noninvasive and convenient for patients compared with wired endoscopy [3,4]. However, due to the location and length of the small bowel, WCE captures images for more than eight hours, generating more than 50,000 images for each examination.

Thus, even an experienced gastroenterologist would need more than two hours to review all of the small bowel images and detect any lesions [5]. This long evaluation time is the main limitation of traditional methods, resulting in high fatigue in physicians. To overcome this limitation and exploit recently developed deep learning methods for automated lesion detection, convolutional neural network (CNN)-based models have been proposed, which have been shown to effectively detect lesions such as bleeding, inflammation, vascular, and polyps among WCE images [6,7,8,9].

Although current CNN-based image-reading models have demonstrated high accuracy in detecting lesions for a given frame, they have limitations in that only one still-shot image is used to detect lesions; thus, they do not benefit from sequential images containing more information. As shown in Figure 1, sequential images offer temporal continuity, which can provide additional information about the lesions. Furthermore, these images possess redundancy, which can help in avoiding false negatives, unlike still-shot images. In the real world, WCE images are stored and displayed as videos, where each frame is captured at an appropriate rate [10]. In addition, if a lesion is suspected during reading, the gastroenterologist not only judges the still-shot image but also refers to its adjacent frames together. Therefore, deep learning models that can consider whole frames of WCE videos are required.

Although CNNs are sufficiently effective at extracting features within a region of an image, they have limited receptive fields, such that their attention is limited to a local area. On the other hand, the recently developed Transformer architecture is known to have advantages over CNNs, particularly in that the Transformer is designed to extract long-term global attention [11]. The Transformer was initially developed for natural language modeling, which achieved significant performance gains over the preceding neural networks. Furthermore, the vision Transformer (ViT), a model that modified the original transformer for computer vision, has also performed well in image classification [12]. Because ViT performs well in computer vision tasks, some studies have employed the transformer architecture to analyze WCE images [13,14,15].

The characteristics of continuous WCE images share significantly more similarities with natural language processing (NLP) tasks than single-image processing tasks. WCE images are not in a single form; instead, they are in a sequential type with continuity. Continuous data have contextual associations. In NLP tasks, the meanings of words are understood differently by other words in the sentence. The recognition process of words is interdependent. In other words, words affect each other. Recurrent neural network (RNN) or long short-term memory (LSTM) architectures are designed to solve problems caused by long-range dependencies by modeling sequential data [16,17]. Similarly, images in video have the same characteristics as words in NLP. In video understanding tasks, other images influence the understanding of one image. For example, if an object that was difficult to distinguish because it was blurry in a specific image is clearly visible in another image and can be distinguished, it becomes possible to accurately distinguish the object even though there is an image in which the object appeared blurry. For this reason, there have been attempts in video tasks to learn by making temporal information into features, such as 3D convolution [18,19,20]. Furthermore, since WCE videos contain relatively many low-quality images compared with other videos, the impact of other images on understanding images is more important. Therefore, it can be expected that the Transformer-based long-range self-attention model would be more effective for WCE video analysis [21].

In this paper, we propose a novel video Transformer model that utilizes the self-attention of images and video information to detect small bowel disease. Unlike the traditional image-level analysis that relies on human-selected images as the input, we conduct a video-level analysis using our proposed video wireless capsule endoscopic network (VWCE-Net). Our code is available at https://github.com/syupoh/VWCE-Net.git (accessed on 7 April 2023). Specifically, our method performs video-level analysis by considering all frames of the video as the input. This model performs video analysis based on spatiotemporal self-attention information obtained from sequential images rather than one still-shot image, as in conventional methods. In the experiments, we compare the proposed method with well-known image-based models such as the YOLOv4 and Xception models [19,20].

## 2. Materials and Methods

### 2.1. Data Acquisition (Data Characteristics)

A total of 260 WCE (MiroCam MC1600, Intromedic Co., Ltd., Seoul, Republic of Korea) cases performed at Dongguk University Ilsan Hospital between 2002 and 2022 were used to train the proposed Transformer model. All WCE image frames are stored in JPEG format with dimensions of 320 × 320 and a frame rate of 3 fps using MiroView 4.0 (Intromedic Co., Ltd., Seoul, Republic of Korea). Our study was conducted with the approval of the Institutional Review Board of Dongguk University Ilsan Hospital (DUIH 2018-10-009-010).

### 2.2. Data Preparation

Two gastroenterologists specializing in capsule endoscopy (Oh DJ and Lim YJ from Dongguk University Ilsan Hospital) independently performed image labeling for lesion detection. After manually reviewing and categorizing the entire WCE case images as normal, bleeding, inflammation, vascular, and polyp tissues, they cross-checked their findings to ensure accuracy. Labels were assigned to each image individually, and a clip is only used for training if all images in the clip have the same label. For example, if a clip contains images with lesions and images without lesions, the clip is not used for training. Therefore, the number of images used for training changes when the clip unit changes. The 40 labeled case datasets are then classified into training (36 cases, 90%) and test (4 cases, 10%) sets. The datasets were composed of clip units rather than still-shot images. Each clip consisted of four sequential images. The reason for setting the unit of one clip as four sequential images is discussed in Section 3.2. The training set consists of 1,291,004 (322,751), 140,788 (35,197), 10,912 (2728), 2328 (582), and 14,832 (3708) images (clips) for normal, bleeding, inflammation, vascular, and polyp tissues, respectively. The test set consists of 172,820 (43,205), 4200 (1050), 304 (76), 892 (223), and 24 (6) images (clips) for normal, bleeding, inflammation, vascular, and polyp tissues, respectively (Table 1).

In previous studies, labeling was traditionally conducted on an individual basis to identify the presence of lesions. However, in this study, we aim to reduce manual labor costs by implementing a simultaneous labeling approach for sequential images, as depicted in Figure 2. Nonetheless, utilizing sequential images as a dataset presents certain limitations. The details of the limitations of the data will be covered in the Discussion section.

### 2.3. Study Design

Our method is implemented using the PyTorch codebase MMAction2 [21]. Figure 3 shows a flow chart of the proposed VWCE-Net. First, the sequential input images are converted into clips comprising several images. Next, the clips are placed into the model and the lesion is detected in each clip.

In this study, we set the clip size to 4 (i.e., four consecutive frames constituted a clip). The size of each frame was 320 pixels in width and height; however, we resized them to 224 pixels. Unlike ViT [12], which takes a single image as input, our method’s input is a clip containing multiple images. Hence, the dimensions of the input are also different from conventional methods, precisely 4 × 224 × 224 × 3 [18], where 4 represents the number of images in the clip. Specifically, the input sequence of VWCE-Net is $x\in R^{4\times 224\times 224\times 3}$ and, as shown in Figure 4, one 224 × 224 image becomes 14×14 (N=196) patches, denoted as $x_{p}\in R^{4\times 196\times 16\times 16\times 3}$, with a patch size of 16×16. These patches are flattened into $x_{fp}\in R^{4\times 196\times 768}$.

BERT is based on the Transformer architecture, which is a powerful language model that can be used to solve a variety of NLP problems, such as question answering, natural language inference, and text classification [22]. In BERT, clstoken is added to the first of all features, used in classification tasks, and ignored in other tasks [22]. After passing through all layers of the Transformer, the “clstoken” acquires the combined meaning of the token sequence. In the classification task, you can pass through this clstoken to the classifier to classify the entire sentence entered. In contrast to BERT, where the input data was in word form, in this work, the input embedding is of the dimension $x_{fp}\in R^{4\times 196\times 768}$, so the clstoken is represented as $t_{cls}\in R^{4\times 1\times 768}.$ The clstoken $t_{cls}$ is concatenated to the flattened patch $x_{fp}$. Finally, the size of the embedding becomes $R^{4\times (196+1)\times 768}$. VWCE-Net learns this clstoken to represent a sequence composed of 196 patched images so that it can operate as a classification token that determines whether there is a lesion or not.

The transformer-based self-attention model does not compute convolution and does not contain recurrence like LSTM. To use the order of sequence information, it is necessary to mathematically model the relative or absolute position of the flatten patches.
(1)$$v_{pos_{\left(t,i\right)}}\left\{\begin{array}{c} \sin\left(t\times w_{k}\right),\;if\;i=2k\;\;\;\;\;\;\;\;\\ \cos\left(t\times w_{k}\right),\;if\;i=2k+1 \end{array}\right.\;w_{k}=\frac{1}{10000^{2k/d}}\;(d\;is\;dimension\;of\;embedding)$$
where t is the position in the sequence and i is the index of the dimension representing the position. Positional encoding $v_{pos}$ has a sinusoidal form. It has a pair value of sine and cosine depending on the value of i. That is, the even-numbered dimension uses sin and the odd-numbered dimension uses cos.
(2)$$v_{pos}=\left[\left(\begin{array}{cccc} v_{pos_{\left(0,0\right)}} & v_{pos_{\left(0,1\right)}} & \cdots & v_{pos_{\left(0,767\right)}}\\ \vdots & \ddots & \ddots & \vdots \\ v_{pos_{\left(196,0\right)}} & v_{pos_{\left(196,1\right)}} & \cdots & v_{pos_{\left(196,767\right)}} \end{array}\right)\right]\;=\left[\left(\begin{array}{cccc} \sin(0) & \cos(0) & \cdots & \sin(0/10000)\\ \sin(1) & \cos(1) & \cdots & \sin(1/10000)\\ \vdots & \ddots & \ddots & \vdots \\ \sin(196) & \cos(196) & \cdots & \sin(196/10000) \end{array}\right)\right]$$

By this positional encoding, in NLP even the same word can have different embedding values depending on the position used in the sentence [11]. In tasks such as text translate and text generation, 1-dimensional positional encoding is calculated because the input is a 1-dimensional word. In the task of this experiment to find a lesion in a given image, since the input is a 2-dimensional image, 2-dimensional position encoding can be considered. To apply 2-dimensional position encoding, first divide embedding in half, set one to X-embedding and the other to Y-embedding. Each size is set to d/2. By concatenating X-embedding and Y-embedding, the final positional encoding value of the patch of the corresponding position can be obtained. This work uses 1-dimensional positional encoding instead of 2-dimensional positional encoding because there is little difference in performance [12]. This means that the 2-dimensional positional relationship information between coordinate of X and Y is sufficiently included in the 1-dimensional positional relationship between the flattened patches.

Then, the embedded feature $z\in R^{4\times 197\times 768}$ is created by adding the position embedding vector $v_{pos}\in R^{4\times 197\times 768}$, which includes the spatial information of the patch. Formally, the embedded feature z is represented as
(3)$$z=\left[t_{cls}||x_{fp}\right]+v_{pos}$$
where || represents the concatenation.

The embedded feature $z\in R^{4\times 197\times 768}$ is projected as query q, key k, and value v representations following the Transformer architecture [11]. This transformation is achieved through linear operations using parameter matrices $W_{qkv}\in R^{768\times (768\times 3)}$, which are described as
(4)$$q,\;k,\;v=zW_{qkv}$$
where the dimensions of the projected q, k, and v are equally 4×197×96. Each operation of q, k, and v contains layer normalization of the embedded feature z.

As shown in Figure 5, the multi-head attention module performs several self-attention operations in parallel. Each of these operations is referred to as a “Head,” following the terminologies in [11]. In this paper, the number of “Heads,” denoted as A, is set to 8. Because the feature dimension is 768, the dimension within the multi-head is 96. Consequently, the q, k, and v vectors are converted into dimensions of 4×8×197×96.

We perform matrix multiplication between q and k, followed by scaling with $D_{h}$, where $D_{h}=D/A$, and then apply the softmax function as [23]
(5)$$a=softmax\left(\frac{q⊗k^{T}}{\sqrt{D_{h}}}\right)$$
which is the self-attention coefficient. In this paper, we set $D_{h}=96.$ Then, the self-attention value is obtained as
(6)s=a⊗v

⊗ denotes element-wise product. Multi-head attention heads are concatenated and passed through the multi-layer perceptron (MLP). For each operation, residual connection is used.
(7)$$z^{\prime }={[s}_{1}|\left|s_{2}||\cdots ||s_{A}\right]+z\;z^{1}=MLP(LN\left(z^{\prime }\right))+z\prime$$

LN denotes layer normalization and MLP is a multi-layer perceptron consisting of two hidden layers. In summary, clstoken $t_{cls}$ is concatenated to the flattened image patch $x_{fp}$ and positional encoding $v_{pos}$ is added to obtain the embedded feature z. Then, z is used to calculate multi-head attention to obtain self-attention s; s passes through MLP to obtain $z^{1}$ in the transformer layer. In this paper, we set the number of transformer layers to 12, so we calculate up to $z^{11}$ by repeating the transformer operation. The first position of $z^{11}$ obtained through the entire Transformer layer is used to determine whether there is a lesion or not.
(8)$$Y=FC(LN\left(z_{\left(0\right)}^{11}\right))$$

FC denotes the fully connected layer $\in R^{768\to 5}$. If the classification result is 0, it means that there is no lesion; if it is more than 1, it means that there is a lesion. The reason why the final output dimension is 5 is because the type of lesion is set to 4 when preparing the data.

### 2.4. Implementation Detail

The basic network that makes the images into patches and passes them through the transformer network is the same as the base ViT architecture [12]. Base ViT was pretrained with ImageNet-21K [24], and in this experiment, 1,291,004 images were additionally trained and fine-tuned. During training, the training images were flipped; the flip ratio was set to 0.5. The momentum was set to 0.9 and the weight decay was set to 0.0001. The learning rate was set to 0.005 and the maximum epoch was set to 100.

To train the YOLOv4 object detection model, we manually labeled bounding boxes around the lesion in the training images. When we evaluated the performance of the model with test images, we evaluated only whether there was a lesion or not. We trained the YOLOv4 object detection model with the following hyperparameters. The momentum was set to 0.949, the weight decay was set to 0.0005, the learning rate was set to 0.001, and maximum iteration was set to 40,000. We used the default hyperparameters from the timm [25] library to train the XceptionNet model. The momentum was set to 0.9, the weight decay was set to 0.0001, the learning rate was set to 0.01, and maximum epoch was set to 100.

## 3. Results

### 3.1. Comparison of Our Video-Level Analysis with Existing Still-Shot Image Analysis Methods

The results of the proposed video classification, image classification, and object detection models are compared in Table 2. The model used for classification was XceptionNet [26], and the model used for object detection was YOLOv4 [27]. XceptionNet is a variant of CNN that is known for its effectiveness in image classification. YOLOv4 is a powerful object detection algorithm that is well suited to real-time applications. It can detect objects quickly and accurately, even in challenging conditions. Existing methods for classification and object detection rely on image-level analysis [19,20], which involves only human-selected images from the entire video. In contrast, we evaluate the entire video in this experiment instead of relying on a human-selected subset.

The sensitivity and specificity performances of the three models are compared with the same full video test dataset. Table 2 shows that the classification model XceptionNet has a low sensitivity rate and that the object detection model YOLOv4 has a low specificity rate compared with the proposed VWCE-Net model.

### 3.2. Determining the Clip Length for the Video-Level Analyses

To investigate the impact of clip length in our video-level analysis, we conducted experiments by dividing the clip length by 2, 4, 6, and 8. The results of these comparisons are presented in Table 3. Surprisingly, we can see that the sensitivity and specificity are not directly proportional to the clip length. Consequently, it is crucial to determine whether to prioritize sensitivity or specificity in determining the optimal clip size. After careful evaluation, we determined a clip size of 4 as being optimal; this clip size yields the highest specificity while maintaining a sufficiently high sensitivity. Therefore, all the explanations previously provided and the subsequent experiments are conducted based on a clip size of 4.

### 3.3. Sampling Strategy

To analyze the WCE images more accurately, sequential images were used as the dataset instead of still-shot images. However, there are limitations to using sequential images as a dataset. Firstly, sequential images often exhibit similarities, particularly between adjacent frames. The abundance of similar images poses a challenge to effective learning and increases the risk of overfitting. Secondly, the dataset suffers from an imbalance, with a substantially larger number of normal images than abnormal ones. This imbalance negatively impacts the model’s performance. To address these challenges, a sampling strategy was employed to select only a portion of the WCE images as clips for training.

When studying a model for action recognition using video, it is common practice to utilize only the clip in the center [28]. However, in this study, in addition to the center sampling of the clip, random sampling was also performed to determine if there was a difference depending on the sampling method. Figure 6 illustrates the scenarios of center sampling and random sampling. A comparison of the random sampling and center sampling approaches is shown in Table 4, which clearly indicates that center sampling yields superior performance.

## 4. Discussion

This study is the first to analyze complete video frames from capsule endoscopy using a transformer network. This approach is motivated by the observation that gastroenterologists typically identify diseases not solely based on individual images but by considering a sequence of images surrounding a suspected area [29]. However, in existing studies that automatically detect symptoms using AI models, the inputs to the system are usually single images as opposed to several consecutive images [19,30,31]. Therefore, it is necessary to manipulate WCE images in a continuous manner to set up an environment similar to actual disease detection.

The transformer is currently a state-of-the-art model for computer vision and NLP tasks. As a result, recent works [13,15] have introduced the application of transformers for processing and analyzing WCE images. These studies have demonstrated the effectiveness of utilizing the transformer architecture in achieving high performance when applied to WCE images.

In previous studies, image-level analysis was performed on data that were selected by a human from the entire video, rather than on the entire video itself. In contrast, our experiment compares the sensitivity and specificity performances of the three models using the same full-video test dataset. As a result, it is shown that the classification model XceptionNet exhibits a low sensitivity rate, whereas the object detection model YOLOv4 exhibits a low specificity rate compared with the proposed VWCE-Net model. It can be seen that our video-based method significantly outperforms the image-based methods in terms of accuracy.

To accurately detect lesions within a clip, we obtained detection results by changing the length of the clip. Additionally, the length of the clip was modified for each result to ensure optimal detection. This is because when the clip is too long, the number of images with lesions in the clip exceeds those without lesions. Hence, to achieve an exemplary detection result, it is crucial to ensure an appropriate clip length that maintains continuity without being too long. As demonstrated in Table 3, it is evident that a clip length of four yields the best results. Accordingly, four images are concatenated into one clip to determine whether there is a lesion in that clip. In addition, considering the considerable length of capsule endoscopy videos, we sample only a portion of the images from the videos for training. Two sampling strategies, center sampling and random sampling, were employed in this process. As illustrated in Figure 6, the video is segmented into sequences whenever the presence or absence of a lesion changes. The center sampling method extracts only the central images from each sequence to form a clip, whereas the random sampling method involves the random extraction of images, irrespective of their position within the sequence.

The center sampling method is widely utilized in video recognition tasks, as it assumes that the images positioned at the center of the sequence are representative of the entire sequence. Conversely, the images extracted through random sampling lack the same level of representativeness. Consequently, incorporating randomly extracted images does not significantly improve the model’s performance. As shown in Table 4, the model trained with randomly extracted images has a lower false positive rate than the model trained with center-extracted images. However, training with center-extracted images yields higher sensitivity.

While randomly extracted images introduce redundant information that restricts the model’s performance improvement, relying solely on center-extracted images is not considered optimal due to the limited information they provide. Therefore, striking a balance between the two sampling methods is crucial to achieving better detection outcomes.

There are some limitations to utilize sequential images as a dataset. Firstly, the presence of numerous similar images within a video sequence poses challenges to effective learning. Learning from a dataset containing numerous similar images increases the risk of overfitting, which can hinder the model’s generalization capability. Secondly, there is a significant data imbalance between normal and abnormal images. The dataset primarily comprises normal images, whereas the number of abnormal images is relatively low. This data imbalance can negatively impact the performance of the model, leading to suboptimal results.

## 5. Conclusions

In this paper, we have proposed a transformer-based neural network for lesion detection in WCE data. Unlike conventional methods that analyze still-shot images, our approach analyzes sets of consecutive images. We trained and tested our network using a dataset of approximately 1.6 million WCE images that were categorized into five classes: normal, bleeding, inflammation, vascular, and polyp. To evaluate the performance of our model, we compared it with existing image-based methods, specifically the XceptionNet and YOLOv4 models. The experimental results clearly demonstrate that our video-based VWCE-Net model outperforms the image-based methods in terms of lesion detection accuracy and effectiveness. By leveraging the power of transformer-based architecture and analyzing WCE data using consecutive image sequences, our proposed approach offers improved performance and holds promise for advancing lesion detection in WCE examinations. To develop the performance of models that handle consecutive image sequences, it is necessary to study automated preprocessing techniques that can remove noise from sequence data before learning. This is because noise in consecutive images can reduce learning efficiency and cause performance degradation in video models that use consecutive images as a single clip. If the automated lesion detection model is further developed, it can be used in practical settings. This would allow gastroenterologists to diagnose patients more efficiently with the help of the automated lesion detection model, reducing their fatigue and allowing more patients to receive high-quality medical care.

## Figures and Tables

**Figure 1 diagnostics-13-03133-f001:**
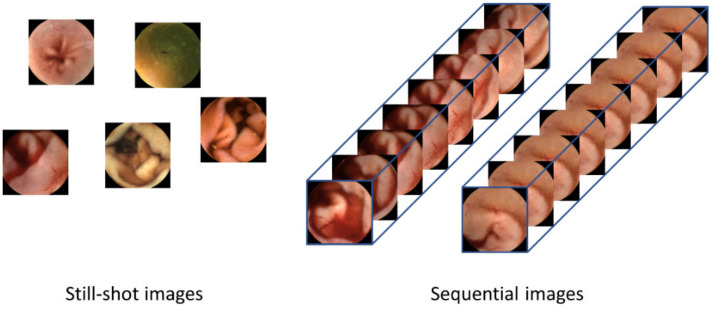
Difference between human-selected still-shot images and sequential images as the inputs to the lesion detection systems. Still-shot images do not have continuity between images, so temporal information cannot be used. Using the whole frames of the video has advantages in that more information can be used and that the temporal changes within the frames can provide valuable cues for detecting lesions.

**Figure 2 diagnostics-13-03133-f002:**
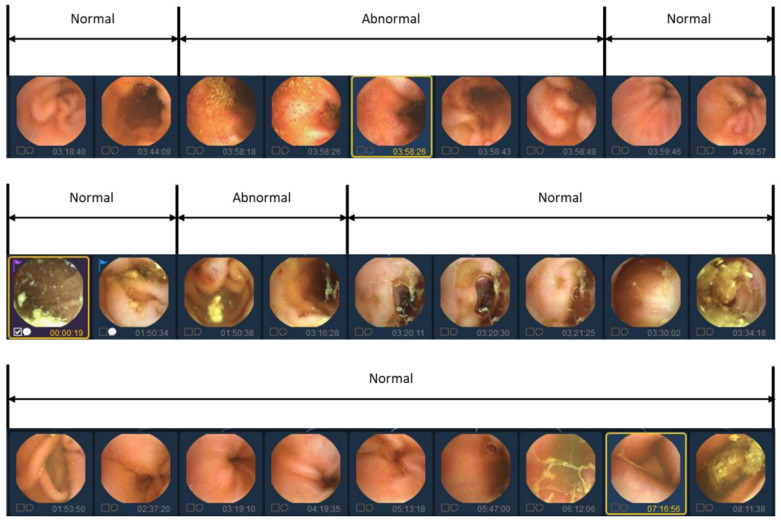
Video data labeling form with abnormal segment display. Because abnormal images in videos tend to occur continuously, they are labeled in sequential images units rather than individual image units.

**Figure 3 diagnostics-13-03133-f003:**
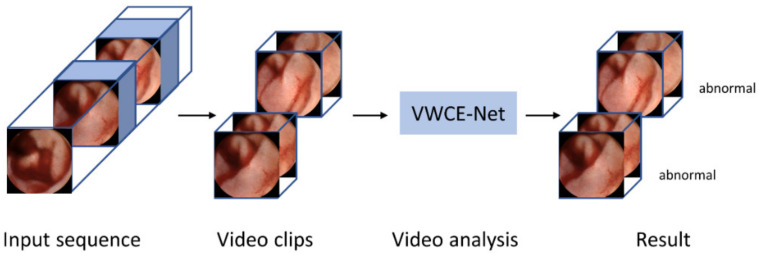
Flowchart of the proposed VWCE-Net. The input sequence is divided into smaller clips and the model then analyzes each clip to determine if it contains a lesion.

**Figure 4 diagnostics-13-03133-f004:**
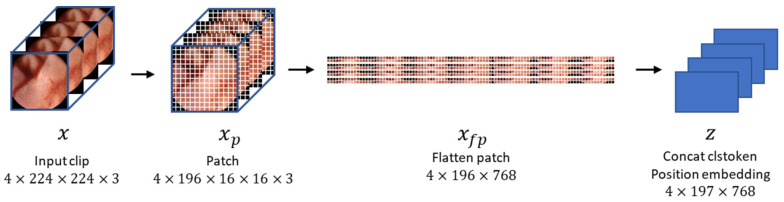
The process by which an input clip becomes an embedded feature. The input clips are preprocessed by dividing them into 16 × 16 patches, flattening them, and concatenating them with the clstoken. The clstoken is a token that detects the presence of lesions. The resulting vector is then used as the feature for the model.

**Figure 5 diagnostics-13-03133-f005:**
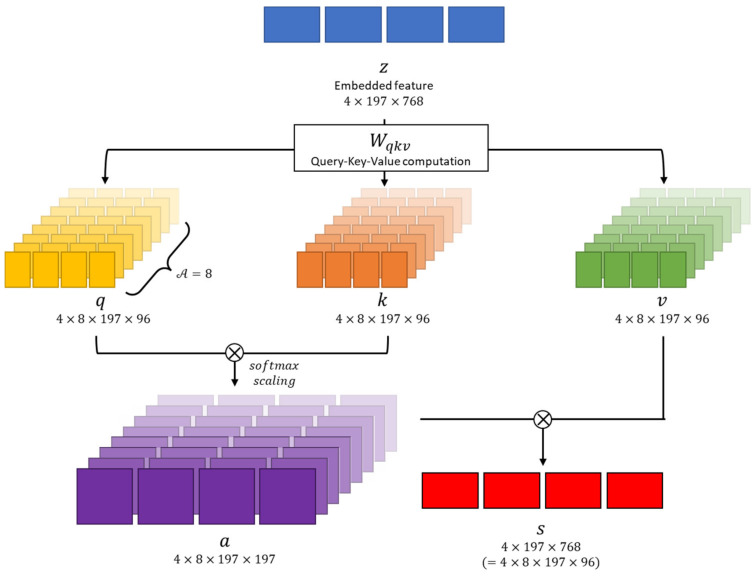
Multi-head attention module overview. This module causes transformer architecture to learn complex relationships between different parts of a sequence.

**Figure 6 diagnostics-13-03133-f006:**
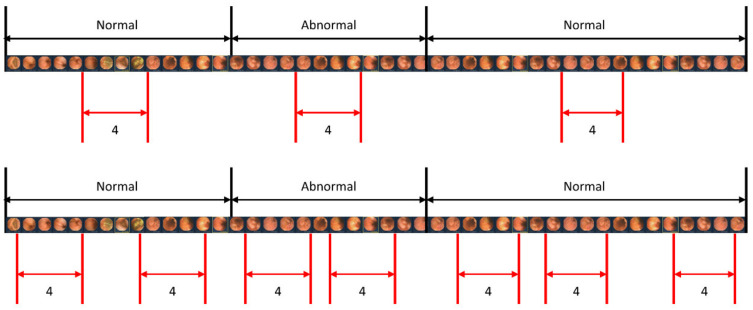
Sampling methods for video input. The upper image represents the center sampling method, where only the center part of the video sequence is extracted as a clip. The lower image demonstrates the random sampling method, where clips are randomly extracted from the entire video sequence.

**Table 1 diagnostics-13-03133-t001:** Dataset specifications.

	Whole Dataset		Training Dataset		Test Dataset	
Classes	Clips	Images	Clips	Images	Clips	Images
Normal	365,956	1,463,824	322,751	1,291,004	43,205	172,820
Bleeding	36,247	144,988	35,197	140,788	1050	4200
Inflammation	2804	11,216	2728	10,912	76	304
Vascular	805	3220	582	2328	223	892
Polyp	3714	14,856	3708	14,832	6	24
Cases		40		36		4

**Table 2 diagnostics-13-03133-t002:** Comparison of the results of the VWCE-Net model based on video analysis with the results of the YOLOV4 and XceptionNet models based on image analysis.

Model	Sensitivity (%)	Specificity (%)
XceptionNet [26]	43.2	90.4
YOLOV4 [27]	88.6	51.5
VWCE-Net	95.1	83.4

**Table 3 diagnostics-13-03133-t003:** Comparative results based on clip length.

Clip Length	Sensitivity (%)	Specificity (%)
2	98.51	72.81
4	95.13	83.43
6	93.93	83.25
8	97.89	80.76

**Table 4 diagnostics-13-03133-t004:** Performance comparison: center sampling vs. random sampling (clip length: 4).

Sampling Strategy	Sensitivity (%)	Specificity (%)
center	95.13	83.43
random	72.84	89.62

## Data Availability

Not applicable.

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
