# Peer review of "Video Analysis of Small Bowel Capsule Endoscopy Using a Transformer Network"

_diagnostics, 2023, doi:10.3390/diagnostics13193133_

Round 1
Reviewer 1 Report
- The manuscript's topic is fascinating and with a growing field of application.
- The introduction provides a good description of the current state of the art in the field of study. However, it would be desirable that, in addition to the data on the precision and usefulness of each methodology, the advantages and the shortcomings of the cited works be commented in greater detail to show more clearly the challenges that remain unsolved.
- The idea of the paper is novel and timely. Authors have identified the issues of existing systems and provided a new mechanism to solve it.
- The code link should be moved to data availability section from abstract.
- In Section propsoed, the authors have shown lots of details of the proposed architecture, but it didn’t be described the architecture well. The description of this architecture is not accordant with code that is used for the implemeantion.
- The proposed method is compared with the other methods, but the experimental settings of the other four methods are not detailed.
- The authors should explain how this study is beneficial for community?
- In Conclusion, future directions and challenges should be explained more.
- How did the authors set the parameters? Please explain why the settings of the parameters are picked.
- Please cite each equation and clearly explain its terms.
- Please polish the English writing and also check the typos errors carefully. In addition, please also make sure the paper format is correct.
- Please polish the English writing and also check the typos errors carefully. In addition, please also make sure the paper format is correct.
Author Response
Manuscript ID: diagnostics-2564445
Type of manuscript: Article
Title: Video analysis of small bowel capsule endoscopy using a Transformer network
Reviewer 1:
- The manuscript's topic is fascinating and with a growing field of application.
- The introduction provides a good description of the current state of the art in the field of study. However, it would be desirable that, in addition to the data on the precision and usefulness of each methodology, the advantages and the shortcomings of the cited works be commented in greater detail to show more clearly the challenges that remain unsolved.
- The idea of the paper is novel and timely. Authors have identified the issues of existing systems and provided a new mechanism to solve it.
- The code link should be moved to data availability section from abstract.
Response: Thank you for kind comments. Existing studies have proven the effectiveness by applying the current deep learning method to lesion detection, but there is a shortcoming that the images are analyzed in one still-shot unit. Therefore, there is a limitation that automated models are greatly affected by noise images when analyzing full-length video. The explanation for this is line 55-65. The code link has been moved to the end of the introduction.
- In Section proposed, the authors have shown lots of details of the proposed architecture, but it didn’t be described the architecture well. The description of this architecture is not accordant with code that is used for the implementation.
Response: Thank you for kind comments. In the Study Design section, a lot of space was devoted to explaining Transformer using attention. Figure 3, figure 4 and figure 5 describe the VWCE-Net structure and Transformer.
- The proposed method is compared with the other methods, but the experimental settings of the other four methods are not detailed.
Response: Thank you for kind comments. We evaluated the performance of three methods: YOLOv4, XceptionNet, and our proposed VWCE-Net. YOLOv4is an object detection model, so we used images with manually bound boxes as training sets. XceptionNet is a classification model, so we used images with assigned classes for each image.
We compared two methods, Yolov4 and XceptionNet, with our proposed VWCE-Net. Since YoloV4 is an object detection model, images that manually bound boxes were used as training sets. Since XceptionNet is a classification model, images assigned class were used for each sheet.
- The authors should explain how this study is beneficial for community?
Response: Thank you for kind comments. We attached an explanation of how this study is beneficial for the community in Conclusion.
After revision : If the automated lesion detection model is further developed, it can be used in practical settings. This would allow gastroenterologists to diagnose patients more efficiently with the help of the automated lesion detection model, reducing their fatigue and allowing more patients to receive high-quality medical care.
- In Conclusion, future directions and challenges should be explained more.
Response: Thank you for kind comments. We added future work for our study in Conclusion.
After revision : To develop the performance of models that handle consecutive image sequences, it is necessary to study automated preprocessing techniques that can remove noise from sequence data before learning. This is because noise in consecutive images can reduce learning efficiency and cause performance degradation in video models that use consecutive images as a single clip.
- How did the authors set the parameters? Please explain why the settings of the parameters are picked.
Response: Thank you for kind comments. We conducted experiments by repeatedly setting parameters in a variety of ways. The parameter setting that achieved the best validation accuracy was selected for the final model.
- Please cite each equation and clearly explain its terms.
Response: Thank you for kind comments. Each meaning of the equations is as follows. (1) and (2) explain the mathematical meaning of positional coding used in the Transformer model. (3) describes the part where clstoken and feature are concatenated and the feature has an attention property by adding the positional encoding value. In (4), the feature was multiplied by matrix to express mathematically the process of dividing into query, key, and value, respectively. (5) is the process of calculating query, key, and scaling factors to obtain an attention value. (6) shows the process of finding a self-attention value by calculating the attention value and the value obtained above. (7) expresses that the process of obtaining the self-attention value is repeated in multi-head attention module. (8) shows the result of determining the presence or absence of lesion with the value obtained from the fully connected layer
- Please polish the English writing and also check the typos errors carefully. In addition, please also make sure the paper format is correct.
Response: Thanks. We checked and corrected the unclear English expressions.
Reviewer 2 Report
This paper proposed a transformer-based neural network for lesion detection in wireless capsule endoscope data. The paper is well-written and easy to follow it. The clinical relevance is clear, the results are convincing. The source codes are publicly available. Remarks:The referencing of Figures and Tables appears to be inconsistent, occasionally cited as "Figure 1" and elsewhere as "Figure 1."
Small typos:
● Page 1:
○ 29: 83.4% sensitivity → 83.4% specificity
○ 35: [1,2] → [1, 2]
○ 40: [3,4] → [3, 4]
● Page 2:
○ 83: [16,17] → [16, 17]
● Page 3:
○ 104: [23,24] → [23, 24]
● Page 5:
○ 144: Figure 2 → Figure 3
● Page 7:
○ 236: [23,24] → [23, 24]
● Page 9:
○ 283: [23,31,32] → [23, 31, 32]
○ 287: [13,15] -> [13, 15]
Introduction:
● 58: reference error
Data Preparation:
● 126: reference error
● Figure 2: The vertical lines exhibit a lack of alignment with the start and end points of the images.
Study design:
● Figure 3: It is necessary to include sentences that provide a description of the visual content represented in the figure.
● Figure 4: It is necessary to include sentences that provide a description of the visual content represented in the figure.
● 159: BERT abbreviation was used without explaining the meaning.
● 176: A separation, or interspace, between the text and the equation is warranted.
● Figure 5: It is necessary to include sentences that provide a description of the visual content represented in the figure.
Implementation detail:
● 227: A reference is requisite for the ImageNet-21K dataset.
Comparison of Our Video-Level Analysis with Existing Still-Shot Image Analysis Methods:
● YOLOv4 and Xception models need an explanation.
● Table 2 should explain the meaning of the used abbreviations in the title (Used abbreviations: ).
● Table 2 should be structured to convey that the values contained within it represent percentages.
Determining the Clip Length for the Video-Level Analyses:
● Table 3 should be structured to convey that the values contained within it represent percentages.
Sampling Strategy:
● Table 4 should be structured to convey that the values contained within it represent percentages.
Discussion:
● 291-303: These sentences bear a significant resemblance to the paragraph found within the results section.
● Page 6:
○ 176: poitional → positional
Author Response
Manuscript ID: diagnostics-2564445
Type of manuscript: Article
Title: Video analysis of small bowel capsule endoscopy using a Transformer network
Reviewer 2:
This paper proposed a transformer-based neural network for lesion detection in wireless capsule endoscope data. The paper is well-written and easy to follow it. The clinical relevance is clear, the results are convincing. The source codes are publicly available. Remarks:
The referencing of Figures and Tables appears to be inconsistent, occasionally cited as "Figure 1" and elsewhere as "Figure 1."
Response: Thank you for kind comments. The referencing of Figures and Tables expressions were consistently unified.
Small typos:
- Page 1:
○ 29: 83.4% sensitivity → 83.4% specificity
○ 35: [1,2] → [1, 2]
○ 40: [3,4] → [3, 4]
- Page 2:
○ 83: [16,17] → [16, 17]
- Page 3:
○ 104: [23,24] → [23, 24]
- Page 5:
○ 144: Figure 2 → Figure 3
- Page 7:
○ 236: [23,24] → [23, 24]
- Page 9:
○ 283: [23,31,32] → [23, 31, 32]
○ 287: [13,15] → [13, 15]
Response: Thank you for kind comments. The typos and reference errors were corrected, and the parts were highlighted. We are grateful for your feedback, which has helped us to make our manuscript stronger.
Introduction:
- 58: reference error
Response: Thank you for kind comments. The reference error in Figure 1 of line 58 was corrected.
Data Preparation:
- 126: reference error
Response: Thank you for kind comments. The reference error in Table 1 of line 126 was corrected.
- Figure 2: The vertical lines exhibit a lack of alignment with the start and end points of the images.
Response: Thank you for kind comments. Figure 2 was revised and newly attached. And a sentence that can better understand Figure 2 was added.
After revision : Figure2 . Video data labeling form with abnormal segment display. Because abnormal images in videos tend to occur continuously, they are labeled in sequential image units rather than individual image units.
Study design:
- Figure 3: It is necessary to include sentences that provide a description of the visual content represented in the figure.
Response: Thank you for kind comments. We added a sentence that can better understand Figure 3.
After revision : Figure 3. Flowchart of the proposed VWCE-Net. The input sequence is divided into smaller clips, and the model then analyzes each clip to determine if it contains a lesion.
- Figure 4: It is necessary to include sentences that provide a description of the visual content represented in the figure.
Response: Thank you for kind comments. We added sentences that can better understand Figure 4.
After revision : Figure 4. The process by which an input clip becomes an embedded feature. The process by which an input clip becomes an embedded feature. The input clips are preprocessed by dividing them into 16x16 patches, flattening them, and concatenating them with the clstoken. The clstoken is a token that detects the presence of lesions. The resulting vector is then used as the feature for the model.
- 159: BERT abbreviation was used without explaining the meaning.
Response: Thank you for kind comments. We added a sentence to explain our BERT and cited the related study.
After revision : BERT is based on the Transformer architecture, which is a powerful language model that can be used to solve a variety of NLP problems, such as question answering, natural language inference, and text classification [22].
- 176: A separation, or interspace, between the text and the equation is warranted.
Response: Thank you for kind comments. We added an interspace between the text and the equation.
- Figure 5: It is necessary to include sentences that provide a description of the visual content represented in the figure.
Response: Thank you for kind comments. We add sentences that can better understand Figure 5.
After revision : Figure 5. Multi-Head Attention module overview. This module makes Transformer architecture to learn complex relationships between different parts of a sequence.
Implementation detail:
- 227: A reference is requisite for the ImageNet-21K dataset.
Response: Thank you for kind comments. We added a reference to ImageNet-21K.
After revision : Base ViT was pretrained with ImageNet-21K [24], and in this experiment, 1,291,004 imag-es were additionally trained and fine-tuned.
Comparison of Our Video-Level Analysis with Existing Still-Shot Image Analysis Methods:
- YOLOv4 and Xception models need an explanation.
Response: Thank you for kind comments. We added a description of the YOLOv4 and XceptionNet models.
After revision : XceptionNet is a variant of CNN that is known for its effectiveness in image classification. YOLOv4 is a powerful object detection algorithm that is well-suited for real-time applications. It can detect objects quickly and accurately, even in challenging conditions.
- Table 2 should explain the meaning of the used abbreviations in the title (Used abbreviations: ).
- Table 2 should be structured to convey that the values contained within it represent percentages.
Response: Thank you for kind comments. The title in Table 2 was changed to have a clear meaning. The meaning of expressing percent was added to Table 2.
After revision : Table 2. Comparison of the results of the VWCE-Net model based on video analysis with the results of the YOLOV4 and XceptionNet models based on image analysis.
Determining the Clip Length for the Video-Level Analyses:
- Table 3 should be structured to convey that the values contained within it represent percentages.
Response: Thank you for kind comments. The meaning of expressing percent was added to Table 3.
After revision : Sensitivity (%), Specificity (%)
Sampling Strategy:
- Table 4 should be structured to convey that the values contained within it represent percentages.
Response: Thank you for kind comments. The meaning of expressing percent was added to Table 4.
After revision : Sensitivity (%), Specificity (%)
Discussion:
- 291-303: These sentences bear a significant resemblance to the paragraph found within the results section.
Response: Thank you for kind comments. We emphasized the importance of video input by changing the corresponding paragraph of the Discussion section.
After revision : In previous studies, image-level analysis is performed on data that is selected by a human from the entire video, rather than on the entire video itself. In contrast, Our exper-iment compares the sensitivity and specificity performances of the three models using the same full video test dataset. As a result, it is shown that the classification model Xcep-tionNet exhibits a low sensitivity rate, while the object detection model YOLOv4 exhibits a low specificity rate compared to the proposed VWCE-Net model. It can be seen that our video-based method significantly outperforms the image-based methods in terms of accuracy.
- Page 6:
○ 176: poitional → positional
Response: Thank you for kind comments. We changed that typo.
Reviewer 3 Report
In this paper, the authors used propose a novel video Transformer model that utilizes the self-at-tention of images and video information to detect small bowel disease. However, there are still a lot of problems in the manuscript.
1. The authors devoted a great deal of space in Materials and Methods to describing the modeling process and the various function formulas. However, Diagnostics is a medical journal, and the authors may need to consider the focus of the manuscript writing.
2. The authors mentioned that two gastroenterologists performed the image labeling, but the dataset was composed of clip units (four sequential images). I would like to know if the label was given for a image or for a clip, and if it was given to the clip, how to deal with 4 consecutive frames containing different classifications?
3. When two experts perform image labeling, how can labels be determined in case of inconsistent cross-checking results? How to categorize or exclude images such as blurring, reflections and food residues?
4. The authors should reconsider the structure of the manuscript, e.g. lines 130-140 "limitations of the dataset" do not seem to fit in this part.
5. The authors compared the three methods in Results part, but did not mention the training and testing related to the other two methods in the Materials and Methods part. In addition, the test performance of the classification and object detection models in this study differed significantly from the results in previously reported studies.
6. There have been many studies reporting the application of AI in capsule endoscopy. Although many studies used selected still images to train the model, they still achieved satisfying performance in the video test, and the authors should discuss the comparison between this study and previous studies.
7. The author may be able to adjust the writing framework and ideas in the discussion part of the manuscript
Author Response
Manuscript ID: diagnostics-2564445
Type of manuscript: Article
Title: Video analysis of small bowel capsule endoscopy using a Transformer network
Reviewer 3:
In this paper, the authors used propose a novel video Transformer model that utilizes the self-at-tention of images and video information to detect small bowel disease. However, there are still a lot of problems in the manuscript.
- The authors devoted a great deal of space in Materials and Methods to describing the modeling process and the various function formulas. However, Diagnostics is a medical journal, and the authors may need to consider the focus of the manuscript writing.
Response: Thank you for kind comments. We are sorry that we have devoted a lot of space to modeling even though it is a medical journal. We also think that the manuscript writing should focus on the medical journal. Therefore, we added an explanation of what benefits the proposed model has in the medical community in Conclusion.
After revision : If the automated lesion detection model is further developed, it can be used in practical settings. This would allow gastroenterologists to diagnose patients more efficiently with the help of the automated lesion detection model, reducing their fatigue and allowing more patients to receive high-quality medical care.
- The authors mentioned that two gastroenterologists performed the image labeling, but the dataset was composed of clip units (four sequential images). I would like to know if the label was given for an image or for a clip, and if it was given to the clip, how to deal with 4 consecutive frames containing different classifications?
Response: Thank you for kind comments. In such a case, a more detailed description of how to determine the label was added.
After revision : Labels were assigned to each image individually, and a clip is only used for training if all images in the clip have the same label. For example, if a clip contains images with lesions and images without lesions, the clip is not used for training. Therefore, the number of images used for training changes when the clip unit changes.
- When two experts perform image labeling, how can labels be determined in case of inconsistent cross-checking results? How to categorize or exclude images such as blurring, reflections and food residues?
Response: Thank you for kind comments. We proceeded by cross-checking, and if opinions were different, we decided together at a meeting. We did not process blurring, reflection, or food residues. We used a clip unit as input to the data to prevent performance degradation caused by noise images.
- The authors should reconsider the structure of the manuscript, e.g. lines 130-140 "limitations of the dataset" do not seem to fit in this part.
Response: Thank you for kind comments. The limitations of the dataset moved to the last part of the discussion. The existing line 130-140 part was modified to be covered in detail in the discussion section.
After revision : The details of the limitations of the data will be covered in the Discussion section.
- The authors compared the three methods in Results part, but did not mention the training and testing related to the other two methods in the Materials and Methods part. In addition, the test performance of the classification and object detection models in this study differed significantly from the results in previously reported studies.
- There have been many studies reporting the application of AI in capsule endoscopy. Although many studies used selected still images to train the model, they still achieved satisfying performance in the video test, and the authors should discuss the comparison between this study and previous studies.
Response: Thank you for kind comments. We added a description of the YOLOv4 and XceptionNet models. In this study, the test dataset is used as the full-length video, not images that are human-selected. Single still-shot image lesion detection models are vulnerable to noise in each of the images. VWCE-Net addresses this weakness by simultaneously analyzing multiple images and using the continuity of multiple images to improve detection performance.
After revision : XceptionNet is a variant of CNN that is known for its effectiveness in image classification. YOLOv4 is a powerful object detection algorithm that is well-suited for real-time applications. It can detect objects quickly and accurately, even in challenging conditions.
- The author may be able to adjust the writing framework and ideas in the discussion part of the manuscript
Response: Thank you for kind comments. The subsections were deleted by modifying the framework of the discussion part.
Round 2
Reviewer 3 Report
The author has made further revisions to improve the manuscript, but I still have some questions
1. As described by the authors, a clip is only used for training if all images in the clip have the same label. This was still an artificial selection of the dataset, essentially. Successive frames of WCE video in a real clinical setting would not contain only the same labels. How would the model determine the classification of multiple lesions in the same field of view?
2. The author mentioned that there was no processing of images such as reflections, so what kind of labeling was given to such images?
3. “There have been many studies reporting the application of AI in capsule endoscopy. Although many studies used selected still images to train the model, they still achieved satisfying performance in the video test, and the authors should discuss the comparison between this study and previous studies.” The author didn't seem to answer my confusion directly. The test set used by the authors was still in fact artificially processed clips, and the performance of the model has not been validated in unprocessed WCE videos, whereas previous studies have validated and achieved a satisfactory level of accuracy.
4. What the authors should have described was the training and testing process of YOLOv4 and XceptionNet in the Methods or Supplementary Material part, instead of just adding a sentence about the models.
Author Response
Reviewer 3:
The author has made further revisions to improve the manuscript, but I still have some questions
- As described by the authors, a clip is only used for training if all images in the clip have the same label. This was still an artificial selection of the dataset, essentially. Successive frames of WCE video in a real clinical setting would not contain only the same labels. How would the model determine the classification of multiple lesions in the same field of view?
Response: Thank you for kind comments. In the training process, only human-selection of clips with the same labels between all images constituting clips is used to form a dataset, so this is artificial selection. However, when testing, the entire full-length video is used as a dataset, so this is not artificial selection. As the reviewer noted, the test dataset clips may not only contain images that match the label. When evaluating, if even one image in the clip has a lesion, the clip should be considered to have a lesion and evaluated.
- The author mentioned that there was no processing of images such as reflections, so what kind of labeling was given to such images?
Response: Thank you for kind comments. When we proceeded with labeling, we performed labeling only based on the presence of lesions. If there was a reflection in the normal image, “Normal” was assigned. If there is a small reflection in an image with a lesion, it is labeled as “Abnormal” without any consideration of reflection. If image analysis was performed by still-shot image, the image would have severely degraded the classification performance of the model.
- “There have been many studies reporting the application of AI in capsule endoscopy. Although many studies used selected still images to train the model, they still achieved satisfying performance in the video test, and the authors should discuss the comparison between this study and previous studies.” The author didn't seem to answer my confusion directly. The test set used by the authors was still in fact artificially processed clips, and the performance of the model has not been validated in unprocessed WCE videos, whereas previous studies have validated and achieved a satisfactory level of accuracy.
Response: Thank you for kind comments. We apologize for not providing a more detailed explanation in our previous response. Like the answer to the first item, in the training step, the clips only use the labels in the clip that match each other. In all images, there are not many clips in which labels in clips do not match each other, so the number of images removed is small. Unlike conventional human-selection, this is a low-cost preprocessing because it is automatically selected rather than removed by humans. The test dataset is not artificially processed because clips are used as test dataset even if the labels in the clip were different. Our study differs from previous studies in that it uses minimal preprocessing of image data during training and evaluates the model on the entire video.
- What the authors should have described was the training and testing process of YOLOv4 and XceptionNet in the Methods or Supplementary Material part, instead of just adding a sentence about the models.
Response: Thank you for kind comments. Following the reviewer's comment, detailed experimental settings of YOLOv4 and XceptionNet were described in the last part of Methods' Implementation Detail section. We are grateful for the helpful feedback that allowed us to improve our manuscript.
After revision: To train the YOLOv4 object detection model, we manually labeled bounding boxes around the lesion in the training images. When we evaluated the performance of the model with test images, we evaluated only whether there was a lesion or not. We trained the YOLOv4 object detection model with the following hyperparameters. The momentum is set to 0.949, the weight decay is set to 0.0005, the learning rate is set to 0.001, and maximum iteration is set to 40000. We used the default hyperparameters from the timm [25] library to train the XceptionNet model. The momentum is set to 0.9, the weight decay is set to 0.0001, the learning rate is set to 0.01, and maximum epoch is set to 100.
Round 3
Reviewer 3 Report
The authors answered my questions in detail and I hope that the model can be further validated in future studies.